# Chemical Characterization of Terpene-Based Hydrophobic Eutectic Solvents and Their Application for Pb(II) Complexation during Solvent Extraction Procedure

**DOI:** 10.3390/molecules29092122

**Published:** 2024-05-03

**Authors:** Mersiha Suljkanović, Jasmin Suljagić, Edita Bjelić, Ante Prkić, Perica Bošković

**Affiliations:** 1Faculty of Natural Sciences and Mathematics, University of Tuzla, Urfeta Vejzagića 4, 75000 Tuzla, Bosnia and Herzegovina; 2Faculty of Technology, University of Tuzla, Urfeta Vejzagića 8, 75000 Tuzla, Bosnia and Herzegovina; jasmin.suljagic@untz.ba (J.S.); edita.bjelic@untz.ba (E.B.); 3Faculty of Chemistry and Technology, University of Split, Ruđera Boškovića 35, 21000 Split, Croatia; prkic@ktf-split.hr; 4Faculty of Science, University of Split, Ruđera Boškovića 33, 21000 Split, Croatia; pboskovic@pmfst.hr

**Keywords:** hydrophobic deep eutectic solvents, chemical characterization, liquid–liquid extraction, Pb(II) ions

## Abstract

Solvents prepared from natural terpenes (menthol and thymol), as H-bond acceptors, and a series of organic acids (chain lengths of 8, 10, and 14 C atoms), as H-bond donors, were characterized and tested as reaction media for liquid–liquid extraction purposes. Due to their high hydrophobicity, they seem to be promising alternatives to conventional (nonpolar and toxic) solvents, since they possess relatively less toxic, less volatile, and consequently, more environmentally friendly characteristics. Assuming that the equilibrium is established between solvent and analyte during a ligandless procedure, it can be concluded that those nonpolar solvents can efficiently extract nonpolar analytes from the aqueous environment. Previous investigations showed a wide range of applications, including their use as solvents in extractions of metal cations, small molecules, and bioactive compounds for food and pharmaceutical applications. In this work, hydrophobic solvents based on natural terpenes, which showed chemical stability and desirable physicochemical and thermal properties, were chosen as potential reaction media in the liquid–liquid extraction (LLE) procedure for Pb(II) removal from aqueous solutions. Low viscosities and high hydrophobicities of prepared solvents were confirmed as desirable properties for their application. Extraction parameters were optimized, and chosen solvents were applied. The results showed satisfactory extraction efficiencies in simple and fast procedures, followed by low solvent consumption. The best results (98%) were obtained by the thymol-based solvent, thymol–decanoic acid (Thy-DecA) 1:1, followed by L-menthol-based solvents: menthol–octanoic acid (Men-OctA) 1:1 with 97% and menthol–decanoic acid (Men-DecA) 1:1 with 94.3% efficiency.

## 1. Introduction

Lead (Pb) is a naturally occurring toxic metal found on earth, that causes environmental pollution and serious public health concerns [1]. It is estimated that more than 75% of lead consumption around the world is for the production of lead–acid batteries specifically for motor vehicles, but also through a variety of industrial processes, such as fuel production, paint production, and construction and in foundries. Water that is transported through lead pipes has the potential to carry lead contaminants [2,3]. Controlling lead in environmental samples is essential due to its toxicity and non-biodegradability [4]. In view of the increasing industrial use of lead and the serious health hazards it poses, there is growing interest in finding new methods for selectively removing, concentrating, and purifying Pb(II) ions from mixtures. In order to remove heavy metal ions from the environment and biological systems, a variety of separation methods have been used including chemical precipitation, ion exchange, membrane filtration, solvent extraction, and liquid membrane transport [5,6,7]. Liquid–liquid extraction (LLE) is a simple, cheap, and versatile technique, making it an interesting separation method for industry. To remove pollutants from aqueous samples, the most commonly used solvents are volatile organic compounds (VOCs) [8]. However, due to their low vapor pressure and high toxicity, it is crucial to replace them with safer ones [9]. Over the past two decades, remarkable advancements have been made in the realm of novel and eco-friendly solvents. The concept of “deep” eutectic solvent (DES) first appeared in the scientific world in 2003 when it was announced that a group of designed solvents could meet the principles of “green” chemistry, unlike the ionic liquids used at the time [10]. Recent studies have shown that deep eutectic solvents (DESs) have immense potential as green substitutes for commonly used toxic organic solvents [11,12]. The majority of synthesized DESs are hydrophilic and form a homogeneous phase in water; however, for some applications, such as extraction from aqueous media, the formation of two-phase systems is required, so HDESs can be used. In a recent study, Sanches et al. reported that hydrophilic DES can lead to eutrophication. Therefore, it is advisable to use hydrophobic DES instead of the hydrophilic variety [13]. In previous studies, Van Osch et al. established four criteria to assess the usability of HDESs; these criteria include low viscosity (less than 100 mPa∙s), high hydrophobicity (low or no water content), density different from water, and minimal influence of pH [14]. Hydrophobic deep eutectic solvents (HDESs), prepared from terpenes like menthol and thymol as well as fatty acids, are considered to be relatively non-toxic, less volatile, more environmentally friendly, and renewable [15,16,17]. Considering their hydrophobicity, they are promising alternatives to conventional organic solvents in sample preparation and LLE of non-polar analytes and transition metals. In this paper, HDESs based on natural ingredients (L-menthol, thymol, and natural organic acids) were prepared and characterized and their effect on the extraction of metal cations was studied. Only chemically stable DESs were selected to be used as solvents in the extraction. This paper aims to examine all factors that affect the possibility of Pb(II) ion removal, by defining the conditions for the extraction of cations from an initial aqueous solution into a hydrophobic solution, all using terpene-based solvents. For the preparation of HDES solvents, L-menthol and thymol were chosen, considering their low viscosity values [18]. The final result is compared with the results of the classical extraction of Pb(II) ions by hydrophobic organic solvents chloroform and 1,2-dichloroethane. Given the relevance of such research and the area of application of hydrophobic eutectic solvents as alternative extractants for heavy metal ions as pollutants still being insufficiently researched, the concept of this research was formulated. The HDESs exhibited favorable characteristics such as low viscosity and high hydrophobicity, which positioned them as promising solvents for extraction from aqueous environments.

## 2. Results and Discussion

### 2.1. Preparation, Stability, and Water Content of HDESs

The preparation of homogeneous liquids immiscible with water was accomplished by mixing solid components. Specifically, L-menthol or thymol were used as hydrogen bond acceptors (HBAs) and were mixed with different hydrogen bond donors (HBDs) in a 1:1 molar ratio. The formation of hydrophobic DESs was investigated using a standard procedure [19]. The first component was weighed in the flask, while the second component was weighed separately and then transferred to the flask [20]. After mixing, the solid components were melted at 40 °C until the resulting solvent was stable [21].

Table 1, shows that HDES solvents containing L-menthol have a water content of approximately 0.025%, while less hydrophobic HDES solvents based on thymol have a water content of 0.05%. The low water content in the hydrophobic DES solvents should not have a significant impact on the density and viscosity values, allowing them to be used in the LLE procedure. The hydrophobic deep eutectic solvents Men-OctA and Men-DecA had the following moisture contents: 0.0258% and 0.0236%, respectively. It was also reported by Florindo et al. that an increase in the alkyl chain of the HBDs resulted in the increased hydrophobicity of the HDESs [15]. For different HDES solvents, water content ranged from 0.02 to 0.11%, so the obtained results were in accordance with the previously established principles for HDES application.

### 2.2. FTIR Characterization

The structure of the prepared natural hydrophobic deep eutectic solvent was confirmed by Fourier Transform Infrared (FTIR) spectroscopy. Figure 1 shows the FTIR spectra for L-menthol, decanoic acid, octanoic acid, and prepared solvents Men-OctA and Men-DecA. In Figure 1, we can observe characteristic bands, among which we can single out stretching vibrations of OH groups (3248 cm^−1^), CH groups (2927.4 cm^−1^ and 2870 cm^−1^), and CO groups (1226 cm^−1^) and bending vibrations of the CH group, typical for menthol (1470 cm^−1^). The broad stretching band of the OH group of menthol (3248 cm^−1^) in HDES solvent Men-OctA was shifted to a higher wavenumber of 3378 cm^−1^ and to 3358 cm^−1^ in the mixture Men-DecA and its intensity decreased. The possible cause for this is the transfer of the electron cloud of O atoms during the formation of the H-bond, which confirmed the existence of the H-bond and the formation of HDESs.

The bands characteristic for CH stretching, which expand in the region between 2650 cm^−1^ and 2950 cm^−1^, show a typical structural rearrangement associated with the loss of the crystal structure of the molecules, due to the phase change that occurred when the two solid components were mixed. By mixing L-menthol and decanoic acid in a molar ratio of 1:1, H-bonds are formed, which is evident in the form of shifting of the characteristic bands (O-H and C=O) in the FTIR spectra of this HDES mixture (Figure 1).

The stretching band of the C=O group is observed at the position of 1710 cm^−1^ in the spectrum of the mixture Men-OctA 1:1, which represents a bathochromic shift to the same band of pure octanoic acid (1706 cm^−1^). Such changes may originate from the influence of nearby electron clouds of menthol, which are spatially located near the carbonyl group of the acid [22].

A shift and decrease in the intensity of the peak originating from the stretching vibration of the carbonyl group of decanoic acid from 1694 cm^−1^ to 1711 cm^−1^ in the Men-DecA 1:1 are observed, indicating the formation of hydrogen bonds. As represented in Figure 2, all thymol-based HDESs also reflected shifts in the initial OH stretching band of their HBA (thymol) from 3177.8 cm^−1^ to 34,178 cm^−1^ in Thy-DecA, while in Thy-OctA, the OH stretching band shifted to 3406.8 cm^−1^. Also, there were shifts from the initial C=O bands in decanoic acid from 1694.2 cm^−1^ to 1707.6 cm^−1^ in Thy-DecA, while Thy-OctA maintained its original C=O band peak but had an increase in its intensity. The results obtained were compared with and validated by previous studies, which showed similar shifts to higher wavelengths for the OH band of the monoterpene and the CO bands of the organic acid, thus depicting interactions between the components, which resulted in the formation of DESs [23,24,25].

### 2.3. Density, Speed of Sound, and Thermal Expansion of HDES

The physical properties of HDESs, including density and sound velocity, are listed in Table 2. The density and sound velocity of HDESs were measured at a temperature range from 293.15 to 313.15 K. In Figure 1, it can be seen that, in the case of all hydrophobic deep eutectic solvents, the densities decrease linearly with temperature, as expected based on [26,27]. Table 3 shows the coefficients of the linear fit, with the values of the standard deviation of the fit. Compared to the density of the thymol-based HDESs in the same temperature range, the density values for the menthol-based solvents are lower. Table 2 represents the effect of temperature on sound velocity u for the studied DESs between 293.15 K and 313.15 K. The speeds of sound decrease linearly with an increase in temperature. The maximum (v) was observed for HDES Thy-DecA (1388.62 ms^−1^) at 298.15 K. The speed of sound (v) decreases in the order of Thy-OctA > Men-DecA > Men-OctA. The results show that the densities of the examined HDES solvents decrease uniformly with increasing temperature. Also, the measured densities are lower than the density of water. The smaller the difference in density between water and the hydrophobic solvent, the more difficult and energy intensive could be the phase separation after extraction. For the most efficient performance of the separation technique, the difference in density between the solvent and water should be as large as possible, which accelerates the process of macroscopic phase separation [28].

Based on the experimentally determined densities, the thermal expansion coefficient *α_p_* can be calculated using the following equation:$$\alpha_{p}=-\frac{1}{d}{\left(\frac{\partial d}{\partial T}\right)}_{p,m}$$
where *α_p_* is the thermal expansion coefficient, *d* is the density of each specially tested deep eutectic solvent, and *T* is the temperature at which the densities and the thermal coefficient are presented. The coefficient of thermal expansion is determined in the same temperature range as the measured density. The coefficient of thermal expansion increases with increasing temperature. By heating in an infinitesimally small interval, there is a faster expansion. The obtained values are given in Table 3.

### 2.4. Viscosity

To use any solvent as a reaction medium, its viscosity has to be considered. Figure 3 shows the viscosities of four different HDESs at temperatures ranging from 293.15 K to 313.15 K. Temperature has a significant effect on the viscosity of solvents. Obtained results show that viscosities decrease with increasing temperature. The higher viscosity values obtained for Men-DecA (21.38 mPa·s at 293.15 K) compared to other solvents indicate that a larger amount of energy is required to initiate viscous flow in the mixture with DecA, which indicates stronger interactions between Men-DecA molecules. The obvious reasons are the weakening of intermolecular interactions and the increase in mobility of DES components [29]. A larger decrease in viscosity was observed at higher temperatures between 298.15 K and 323.15 K. All of the solvents that were examined met the requirement of having viscosity values below 100 mPa∙s.

### 2.5. Application of Selected Solvents in LLE Procedure

To verify whether the selected hydrophobic DESs are capable of extracting metal ions from the aqueous solutions, this study employed model systems that contained buffered aqueous solutions of Pb(II) ions (accompanied with coexisting cations), as well as hydrophobic solvents prepared prior to experiments. The mechanism and factors that affect the selective extraction of Pb(II) ions were investigated, i.e., the type and volume of solvent used, the concentration of coexisting ions in the feed solution, and the equilibrium time of extraction. Treatment of model systems with solvent volumes less than 2 mL resulted in significantly reduced extraction efficiency compared to treatment of systems with DES volumes of 5 mL, carried out under the same conditions. Considering that concentration gradients are driving forces in mass transfer, these results are consistent with the principles of mass transfer. The extraction efficiencies were slightly lower but still higher than 80% when volumes of 3 mL and 4 mL of solvent were used. Considering that reducing the amount of solvent did not significantly change the amount of removed ions, 3 mL was chosen as the optimal parameter for further experiments. To emphasize the advantage of the proposed method for the extraction of Pb(II) ions, we make a comparison with the results of conventional liquid–liquid extraction with chloroform (CH) and 1,2-dichloroethane (DCE) as solvents. The results obtained from this research are shown in Figure 4. A comparison was made between the results obtained using the HDES solvents and with equimolar concentrations of analyte ions and counter ions within the feed solution, as well as in the conventional LLE procedure with CH and DCE solvents. However, to obtain satisfactory extraction efficiency, conventional hydrophobic solvents required 120 min of mixing at 300 rpm, and comparable results for HDESs were achieved after only 15 min. Actually, the highest extraction efficiency for Thy-DecA solvent (98%) was achieved, compared to other HDESs as well as CH and DCE. The extraction efficiency rate decreases in the order of Men-OctA (97%) > Men-DecA (94.3%) > Thy-OctA (90.3%). In the back-extraction procedure, we used ethylenediaminetetraacetic acid (EDTA) as a stripping agent. EDTA has a strong complexing ability with Pb(II) ions. Back-extraction efficiency was higher for all HDESs used compared to conventional solvents.

The study investigated the impact of equimolar concentrations of Ni(II), Co(II), Zn(II), Mn(II), and Cd(II) ions on the efficiency of extraction and back-extraction of Pb(II) ions. The results showed that the presence of investigated ions did not interfere with the removal of Pb(II) ions. In the conventional LLE procedure, ligands are necessary as “carriers” of metal ions. In the proposed procedure with HDESs as solvents, the use of macrocyclic ligands is not required, since the analyte ions are transferred into the solvent directly, without ligands as mediators.

## 3. Materials and Methods

### 3.1. Solvents, Solutions, and Reagents

Pb(II), Ni(II), Co(II), Zn(II), Mn(II), and Cd(II) solutions were prepared by dilution, starting from the stock solutions (1000 mg L^−1^) Merck (Darmstadt, Germany) to achieve the intended concentrations. Standard metal solutions used for calibration of the AAS spectrophotometer were prepared by diluting their respective 1000 mg L^−1^ stock solutions. For HDES preparation, L(-) menthol (CAS number: 2216-51-5; mass fraction purity: ≥99%), thymol (CAS number: 89-83-8; mass fraction purity: ≥99%), octanoic acid (CAS number: 124-07-2; mass fraction purity: ≥99%), dodecanoic acid (CAS number: 143-07-7; mass fraction purity: ≥99%), were obtained from Acros Organics (Antwerpen, Belgium) and decanoic acid (CAS number: 334-48-5; mass fraction purity: ≥99%) was procured from Alfa Aesar (Ward Hill, MA, USA). Disodium EDTA (CAS number: 6381-92-6; mass fraction purity: ≥99%) were obtained from Sigma-Aldrich (St. Louis, MO, USA).

### 3.2. Characterization Methods

The densities and sound velocities of HDESs were measured at a pressure of 101.3 KPa using a DSA 5000 M density and velocity meter from Anton Paar (Graz, Austria). The instrument has a reproducibility of 0.000005 g/cm^3^. The viscosities of the DESs were measured using an Anton Paar Lovis 2000 M/ME viscometer (Graz, Austria), which had a relative accuracy of 0.005 and was set to a pressure of 101.3 kPa. To obtain the average values, the viscosities were calculated three times. The water content of the individual HDES was measured by Karl Fischer titration (Mettler Toledo DL38 Volumetric KF Titrator) (Columbus, OH, USA). A PerkinElmer Spectrum Two FTIR spectrometer was used to analyze HDESs. The spectrometer used infrared light in the range of 500 to 4000 cm^−1^ to obtain the spectra of the samples.

Quantification of metal ions in the solvent extraction experiments was obtained by Flame Atomic Absorption Spectrometry technique, using the instrument Perkin Elmer AAnalyst 200 (Norwalk, CT, USA).

### 3.3. Extraction Procedure

For the optimized liquid–liquid extraction procedure, 5 mL of feed solutions containing the analyte (Pb(II) solution, 1 × 10^−4^ mol/L^−1^) and counter ions (picrates, 1 × 10^−3^ mol/L^−1^) were mixed with the appropriate amount of the chosen solvent (different volumes in the range of 1–5 mL were tested). Extraction efficiency was determined after 15 min of mixing at 300 rpm, after which the two phases of different polarities were physically separated. In the aqueous phase, the quantification of analyte ions was performed by FAAS, using the calibration curve method. For the back-extraction procedure, 5 mL of a buffered aqueous solution of the stripping agent, EDTA, at 1 × 10^−3^ mol/L^−1^, was used.

## 4. Conclusions

Four hydrophobic deep eutectic solvents from thymol and L- menthol and long-chain organic acids were successfully prepared, all in molar ratios of 1:1. Physicochemical analysis was carried out on the prepared HDESs, which revealed their excellent characteristics. The formation of hydrogen bonds was confirmed by infrared spectroscopy. Since most HDES solvents are prepared from natural raw materials, the solvents are considered relatively non-toxic, environmentally friendly, and sustainable. The densities of investigated HDESs were within the range of 0.885071–0.943114 g/cm^−3^; all densities were found to decrease with an increase in temperature. The viscosities of the HDESs were found to be lower than 25 mPa·s and decreased with an increase in temperature. The use of L-menthol and thymol HDES solvents in the technique of liquid–liquid extraction of Pb(II) ions showed many advantages: simplicity and shorter duration of the procedure, less toxicity of the solvent, and high efficiency of analyte ion removal. The most efficient extraction of Pb(II) ions is achieved using 3 mL of HDES solvent Thy-DecA (1:1), without counter ions in the feed solution. The higher efficiency of HDES solvent extraction gives this procedure an advantage over the conventional ones.

## Figures and Tables

**Figure 1 molecules-29-02122-f001:**
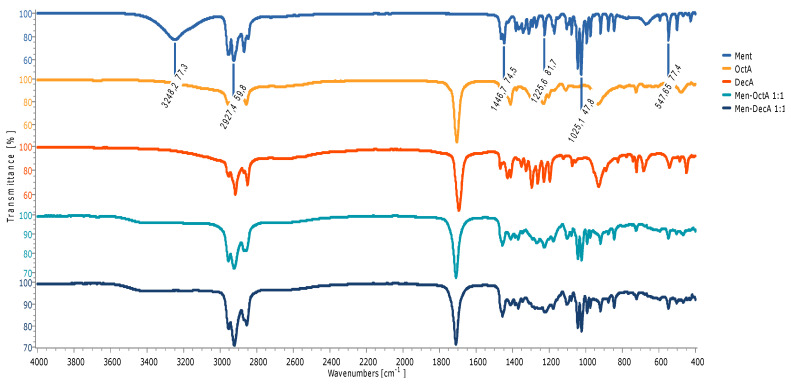
FTIR spectra of individual components and HDESs based on L-menthol.

**Figure 2 molecules-29-02122-f002:**
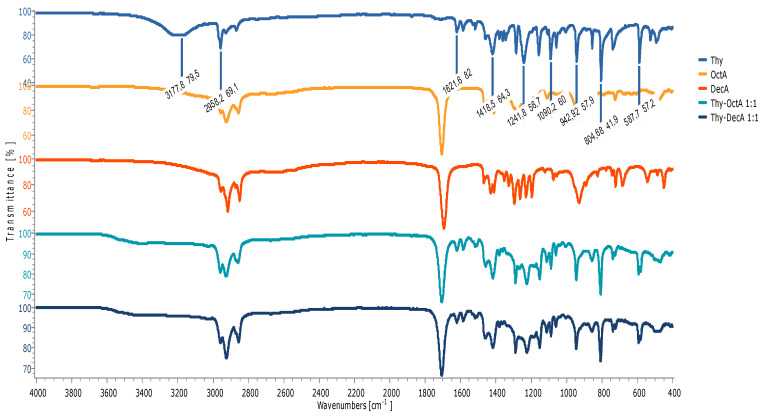
FTIR spectra of individual components and HDESs based on thymol.

**Figure 3 molecules-29-02122-f003:**
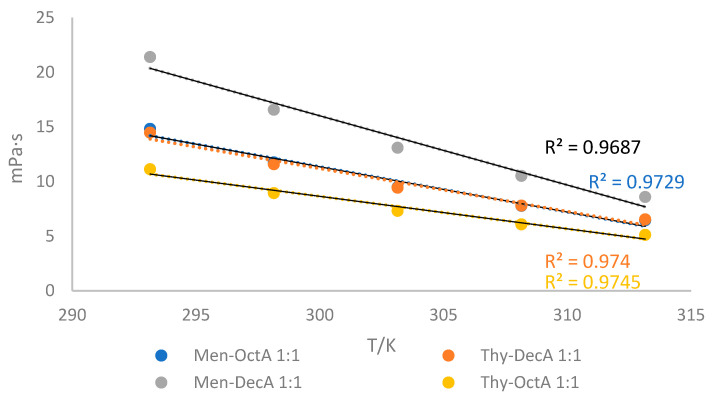
Viscosity values (*η*) as a function of temperatures in interval T = 293.15–313.15 K for investigated hydrophobic deep eutectic solvents.

**Figure 4 molecules-29-02122-f004:**
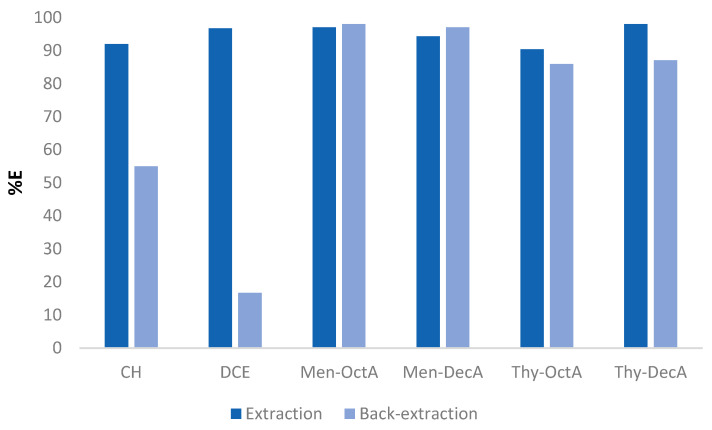
Comparison of extraction/back-extraction efficiency in procedures with HDES solvents and classic chlorinated organic solvents.

**Table 1 molecules-29-02122-t001:** Composition, abbreviations, and water content of prepared HDESs.

HBA	HBD	Composition	Abbreviations	Water Content/w %
Menthol	Octanoic acid	1:1	Men-OctA	0.0258
	Decanoic acid		Men-DecA	0.0236
Thymol	Octanoic acid	1:1	Thy-OctA	0.05
	Decanoic acid		Thy-DecA	0.049

**Table 2 molecules-29-02122-t002:** Measured density and sound velocity values for HDESs.

T (K)	HDES							
	Density d (g/mL)				Sound Velocity (ms^−1^)			
	Men-OctA	Men-DecA	Thy-OctA	Thy-DecA	Men-OctA	Men-DecA	Thy-OctA	Thy-DecA
293.15	0.905008	0.899754	0.943114	0.939875	1352.07	1369.82	1379.33	1388.76
298.15	0.901257	0.896093	0.939164	0.927024	1334.92	1353.03	1362.37	1371.84
303.15	0.897506	0.892429	0.935212	0.923173	1317.89	1336.11	1345.4	1354.96
308.15	0.893748	0.888753	0.931259	0.919322	1300.94	1319.10	1328.48	1338.10
313.15	0.88998	0.885071	0.927296	0.915466	1283.95	1302.40	1311.64	1321.32

**Table 3 molecules-29-02122-t003:** Thermal expansion coefficient (*α_p_*) of different deep eutectic mixtures in the temperature range from 293.15 to 313.15 K and at atmospheric pressure (*p* = 0.1 MPa).

	HDES			
T (K)	α∙104 (K^−1^)			
	Men-OctA	Men-DeA	Thy-OctA	Thy-DecA
293.15	8.30	8.16	8.39	12.99
298.15	8.34	8.19	8.42	13.17
303.15	8.41	8.23	8.46	13.22
308.15	8.50	8.26	8.49	13.28
313.15	8.44	8.29	8.52	13.33

## Data Availability

Data supporting this study are included within the article.

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
