# Peer review of "Chemical Characterization of Terpene-Based Hydrophobic Eutectic Solvents and Their Application for Pb(II) Complexation during Solvent Extraction Procedure"

_molecules, 2024, doi:10.3390/molecules29092122_

Round 1
Reviewer 1 Report
Comments and Suggestions for Authors The manuscript entitled "Physicochemical characterization of terpene-based hydropho-bic eutectic solvents and their application as reaction media for Pb(II) complexation during LLE procedure" describes the characterization and the use of DES as a means of Pb extraction from the aqueous phase. Although the subject matter is interesting, the paper is at an immature stage and may not be published until it has been thoroughly revised. The context regarding lead as a pollutant is completely missing in the introductory section. In particular, it is not clear why lead was chosen among the various heavy metals. Furthermore, the current extraction methods and how the use of DES is better than the state of the art are not reported. The lack of comparison with the literature is also missing throughout the manuscript in the characterization part of the DESs. The investigated DESs are not new and a comparison should also be made taking into account the different water content (e.g. 10.1016/j.cej.2021.128472, ........). DESs that do not result in the formation of a clear and homogeneous phase should be excluded from the manuscript as they are not investigated in any type. A different molar ratio should have been investigated for them, since DES are such in all compositions (the definition of Martins et al. should be added, 10.1007/s10953-018-0793-1). Recent studies show how the properties of DES can be modulated by changing the molar ratio (10.1016/j.molliq.2023.121563). Another aspect worth highlighting is how hydrophobic DESs may be better than hydrophilic DESs. Vieira Sanches et al. (10.1007/s11356-022-23362-5) report how hydrophilic DES can lead to eutrophication. This may not be possible with hydrophobic DES and is an excellent strength to highlight. Finally, I suggest deleting "as a reaction medium" in the title, as the result does not suggest that a reaction takes place in the DES, only an extraction. Minor The units i should be separated from the value by a space. Constants should be italicized in the text. In the 4th line of the results, it is not clear what the authors mean by "on a scale". Table 1 is out of position In Figures 1 and 2, the scales and axis labels are too small.In the caption of Figure 3, it is not clear why the temperatures are in brackets
Comments on the Quality of English Language
English should be improved.
Author Response
Please see the attachment. On the other hand, we have decided to use MDPI language service for English editing.

Reviewer 2 Report
Comments and Suggestions for Authors
The authors present a work on Physicochemical characterization of terpene-based hydrophobic eutectic solvents and their application as a reaction media for Pb(II) complexation during LLE procedure
1-First of all, the title could not explain the concepts of the manuscript contents for readers; based on the concept of the manuscript, as instance “Physicochemical” word in the title is not acceptable due to that there is not any significant concept of physical phenomenon. The authors should use only “chemical” word instead of “physicchemical”
2- In many places in the text the abbreviation does not defined such as “LIE”, and some of the abbreviation is not defined in the proper places, which should be moved in the text
3- The title of table 1 was written alone without any table
4- In one of the tables the abbreviations for Octanoic acid, Decanoic acid, is mentioned, 2-hydroxybenzoic acid, and Tetra-decanoic acid are written, Men-OctA, Men-DecA, Men-SalA, and Men-tDecA , which seems is not complete. What does mean “Men” in all of them, more explanation is needed.
5-There is a table in the text without any title
6- The abbreviation {HDESs of Hydrophobic deep eutectic solvents}, should be appears in the first apply it in the text, not in the middle of manuscript, while several HDESs in the first part of manuscript have been used without any definition
7-Although EDTA is known for some of scientist, it should be defined as Ethylene di-amine tetra acetic acid in the text.
8- In total in the text, important words and abbreviation is not written and a major revision for these words should be checked carefully again
9-The only unique formula in the text is a copy and it should be type by the word, in addition due to only it is one formula the phrase (1) is not necessary
10- In FTIR characterization in figure 1 all of the peaks should be define, Peaks without any definition are not acceptable
11- I can’t understand how the authors calculated such these accurate data for speed of sound. More explanation are needed
12- Keywords are not comprehensive, due to this fact that this work is based on reaction media
for Pb(II) complexation during LLE procedure. As instance the key words “LLE” should be added or natural terpenes is not important keyword for this work. A major revision in keywords based on concept of manuscript are needed
13- The authors does not discussed on the amounts of tables in the text. They have to discuss more about these data
14- The correlation coefficient and regression of the line in figure 3 should be define
15- Introduction cannot explain the aim of manuscript clearly. As instance this paragraph is not related to the introduction .”In this paper, hydrophobic deep eutectic solvents based on natural ingredients (L-menthol, Thymol, and natural organic acids) were prepared and characterized and their effect on the extraction of metal cations was studied”.
16- Results is belong to section 2 (2.results) and then section 3 is 3.Materials and Methods. Where is the section related to discussion? Discussion is the most important parts for any papers
17. The text of conclusion is like an experimental section results. It should be revised
18. The number of references is not enough (most of references are old), more references from recent years (2022 and 2023) are needed.
19. What we can learn from figures 3 and 4? The explanations of authors are not sufficient.
20. What we learn from table 1, more explanation is needed
In my opinion the manuscript is not prepared well and written in unacceptable designing and it is difficult for understanding, therefore it can be accepted by a major revision
Comments on the Quality of English LanguageMinor editing of English language required
Round 2
Reviewer 1 Report
Comments and Suggestions for Authors
The authors responded to my remarks. Therefore, I recommend this work for publication.
Comments on the Quality of English LanguageEnglish level is satisfactory.
Reviewer 2 Report
Comments and Suggestions for Authors
Minor editing of English language required
Comments on the Quality of English LanguageAccept in present form